# Clonal hematopoiesis is associated with adverse outcomes in multiple myeloma patients undergoing transplant

Tarek H. Mouhieddine [1,2,3], Adam S. Sperling[1,2], Robert Redd [4], Jihye Park[1,3], Matthew Leventhal[3], Christopher J. Gibson[1], Salomon Manier[1,5,6], Amin H. Nassar [1,7], Marzia Capelletti[1], Daisy Huynh[1], Mark Bustoros [1,2,3], Romanos Sklavenitis-Pistofidis[1,2,3], Sabrin Tahri[1,2,3], Kalvis Hornburg[1], Henry Dumke[1], Muhieddine M. Itani[8], Cody J. Boehner[1], Chia-Jen Liu [1], Saud H. AlDubayan[1], Brendan Reardon[1], Eliezer M. Van Allen[1], Jonathan J. Keats [9], Chip Stewart[4], Shaadi Mehr[10], Daniel Auclair[10], Robert L. Schlossman[1], Nikhil C. Munshi[1], Kenneth C. Anderson[1], David P. Steensma [1], Jacob P. Laubach[1], Paul G. Richardson[1], Jerome Ritz [1], Benjamin L. Ebert [1,2,3], Robert J. Soiffer[1], Lorenzo Trippa[1,11], Gad Getz [3,8], Donna S. Neuberg[4] & Irene M. Ghobrial [1,2,3✉]

Multiple myeloma (MM) is a plasma-cell neoplasm that is treated with high-dose chemotherapy, autologous stem cell transplant (ASCT) and long-term immunomodulatory drug (IMiD) maintenance. The presence of somatic mutations in the peripheral blood is termed clonal hematopoiesis of indeterminate potential (CHIP) and is associated with adverse outcomes. Targeted sequencing of the stem cell product from 629 MM patients treated by ASCT at the Dana-Farber Cancer Institute (2003–2011) detects CHIP in 136/629 patients (21.6%). The most commonly mutated genes are DNMT3A, TET2, TP53, ASXL1 and PPM1D. Twenty-one from fifty-six patients (3.3%) receiving first-line IMiD maintenance develop a therapy-related myeloid neoplasm (TMN). However, regardless of CHIP status, the use of IMiD maintenance associates with improved PFS and OS. In those not receiving IMiD maintenance, CHIP is associated with decreased overall survival (OS) (HR:1.34, $p = 0.02$) and progression free survival (PFS) (HR:1.45, $p < 0.001$) due to an increase in MM progression.

[1] Department of Medical Oncology, Dana-Farber Cancer Institute, Boston, MA 02115, USA. [2] Harvard Medical School, Boston, MA 02115, USA. [3] Broad Institute of MIT and Harvard, Cambridge, MA 02142, USA. [4] Department of Data Sciences, Dana-Farber Cancer Institute, Boston, MA 02115, USA. [5] Department of Hematology, CHU, Univ. Lille, 59000 Lille, France. [6] INSERM UMR-S1172, 59000 Lille, France. [7] Department of Medicine, Brigham and Women's Hospital, Boston, MA 02115, USA. [8] Massachusetts General Hospital, Boston, MA 02114, USA. [9] Integrated Cancer Genomics Division, Translational Genomics Research Institute, Phoenix, AZ 85004, USA. [10] Multiple Myeloma Research Foundation, Norwalk, CT 06851, USA. [11] Harvard T.H. Chan School of Public Health, Boston, MA 02115, USA. ✉email: irene_ghobrial@dfci.harvard.edu

Multiple myeloma (MM) is an incurable clonal plasma-cell malignancy that is more common in older individuals and accounts for approximately 10% of all hematological malignancies[1]. The standard of care for fit newly diagnosed patients in the United States is induction chemotherapy with a combination of a proteasome inhibitor and immunomodulatory drug (IMiD) followed by high dose melphalan chemotherapy and autologous stem cell transplantation (ASCT). This is typically followed by IMiD maintenance until disease progression. Not all patients benefit from ASCT and it can be associated with significant long-term toxicities including cytopenias and therapy-related myeloid neoplasms (TMN), a risk that is further increased by the use of lenalidomide maintenance[2–4]. However, the clinical benefit of lenalidomide maintenance as seen in improved overall survival (OS) clearly outweigh its risk. A better understanding of which patients may benefit from ASCT and maintenance therapy is an important question and active area of ongoing clinical investigation.

A number of recent studies have identified recurrent somatic mutations in the blood of otherwise healthy adults, a condition referred to as clonal hematopoiesis of indeterminate potential (CHIP)[5]. CHIP is associated with a 0.5–1% risk of progression to a non-plasma-cell hematologic neoplasm, in particular myelodysplastic syndrome (MDS) and acute myeloid leukemia (AML)[6–8], a situation analogous to monoclonal gammopathy of undetermined significance (MGUS). CHIP is also associated with higher all-cause mortality largely mediated by increased risk of cardiovascular disease, myocardial infarction and stroke[8–11].

In these initial large studies, the prevalence of CHIP was found to increase with age, with a prevalence of approximately 1% in individuals under the age of 50 but 10–15% in those older than 70[7,8,12]. Because these studies used whole genome or whole exome sequencing, they had a limited ability to identify small clones with a variant allele fraction (VAF) less than 0.02. More recent studies utilizing molecular barcodes and deep targeted sequencing have identified somatic mutations at VAFs as low as 0.0001, demonstrating that small clones can be detected in virtually all adults[13]. However, the clinical significance of these small clones is unclear as the presence of mutations at a VAF greater than 0.01 carried the strongest association with the development of a subsequent myeloid malignancy[13–17]. Additional work is ongoing to identify clinically meaningful VAF cutoffs, which also may vary depending upon the specific clinical scenario and the patient outcomes being measured.

The prevalence of CHIP is higher in patients with cancer who have been exposed to cytotoxic chemotherapy or radiation and is associated with worse clinical outcomes, including a higher risk of progression to a TMN[18]. Multiple reports have also demonstrated that CHIP is detectable in up to 30% of patients with non-Hodgkin lymphoma (NHL) at the time of ASCT and is associated with increased risk of TMN and non-relapse mortality[19–21]. The presence of CHIP has been reported in patients with MM, but that study was not powered to assess a relationship between CHIP and clinical outcomes[22].

In this study (see Supplementary Fig. 1), we investigate the prevalence of CHIP at the time of ASCT in MM patients and find that it confers worse clinical outcomes in relation to a faster MM progression rate but does not pose an increased risk of TMN. We also show that IMiD maintenance therapy improves outcomes irrespective of CHIP status.

## Results

### CHIP mutational spectrum.

To determine the mutational spectrum and prevalence of CHIP in patients with MM we performed targeted sequencing of 224 genes recurrently mutated in hematologic malignancies on DNA purified from mobilized stem cell products collected prior to ASCT (Supplementary Tables 1 and 2). CHIP is defined as the presence of characteristic leukemia-associated somatic mutations in hematopoietic cells that occur at a VAF of at least 0.02 in the absence of diagnostic criteria for hematological neoplasms[9]. Accordingly, we detected somatic mutations in 88 individuals (14%), and that number rose to 136 (21.6%) patients when including smaller clones with a VAF of 0.01 or higher. The median VAF was 0.027 and 24/136 patients (17.6%) had a clone with VAF ≥ 0.10 (Supplementary Fig. 2). *DNMT3A* was the most frequently mutated gene followed by *TET2*, *TP53*, *ASXL1* and *PPM1D* (Fig. 1a, Supplementary Table 3). We next sought to verify that the detected somatic mutations were not coming from circulating MM cells, but tumor samples were not available for sequencing. However, with the exception of *TP53*, these genes are not recurrently mutated in MM[23–27] and prior work has demonstrated that plasma cell contamination of peripheral blood (PB) stem product is generally minimal[28,29]. In order to confirm this, we performed ultra-low-pass whole-genome sequencing (ULP-WGS) on the stem cell products to detect large-scale copy number alterations, which reflect tumor cells. Only 49 patients had a measurable MM tumor fraction (3–5.4%) in their stem cell product and removal of these samples from statistical analysis did not alter our results.

A single CHIP mutation was detected in 116/136 patients (85.3%), while 20 patients (14.7%) had 2 or more mutations in different genes (Fig. 1b, c). Consistent with prior studies[7,8], 65% of *DNMT3A* mutations were truncating and 35% were missense, with the p.R882 mutation constituting 45% of all *DNMT3A* missense mutations (Supplementary Table 4). The *ASXL1* and *PPM1D* mutations were all truncating, while 80% of *TET2* mutations were truncating and 91% of *TP53* mutations were missense. Collectively, these data demonstrate that CHIP is common in patients with MM. *TP53* mutations appear to be somewhat enriched in our population, but the overall mutational spectrum is more similar to that seen in healthy individuals in the general population than to patients with NHL undergoing ASCT where mutations in *PPM1D* and *TP53* predominate.

### Clinical associations with CHIP.

The median age of all patients in our cohort was 58 years (range: 24–83) at time of ASCT with median follow-up of 9.7 years (range: 0.2–16.6) post-ASCT. Consistent with prior reports[7,8], we found that the presence of CHIP was associated with aging ($p < 0.001$), reaching a prevalence of 47% in patients over the age of 70 (Table 1). The median age at ASCT in patients with CHIP was 61 years, compared to 57 years in those without CHIP ($p < 0.001$). The presence of CHIP was associated with a history of smoking ($p = 0.04$), past medical history of cancer ($p = 0.04$) and a decreased efficiency of stem cell mobilization (5.8 million CD34+ cells/kg/day in those with CHIP compared to 8.3 million CD34+ cells/kg/day in those without CHIP ($p = 0.03$) (Table 1). Finally, the presence of CHIP was not associated with abnormalities in the PB parameters at diagnosis or post-induction (Supplementary Figs. 3 and 4).

### Risk of TMN.

The presence of CHIP is associated with an increased risk of AML and MDS in healthy individuals and TMN in those undergoing ASCT for relapsed NHL[8,19]. We therefore sought to determine whether this was the case in patients with MM. During follow-up, 21 patients developed MDS or AML, diagnosed at a median of 4.4 years post-ASCT (range: 0.9–9.9). All 21 patients who developed MDS/AML received maintenance with thalidomide or lenalidomide, for a median of 2.69 years (range: 0.45–5.48) before developing a TMN (16 first-line and 5 post-relapse). Unlike in the setting of NHL[19], the presence of

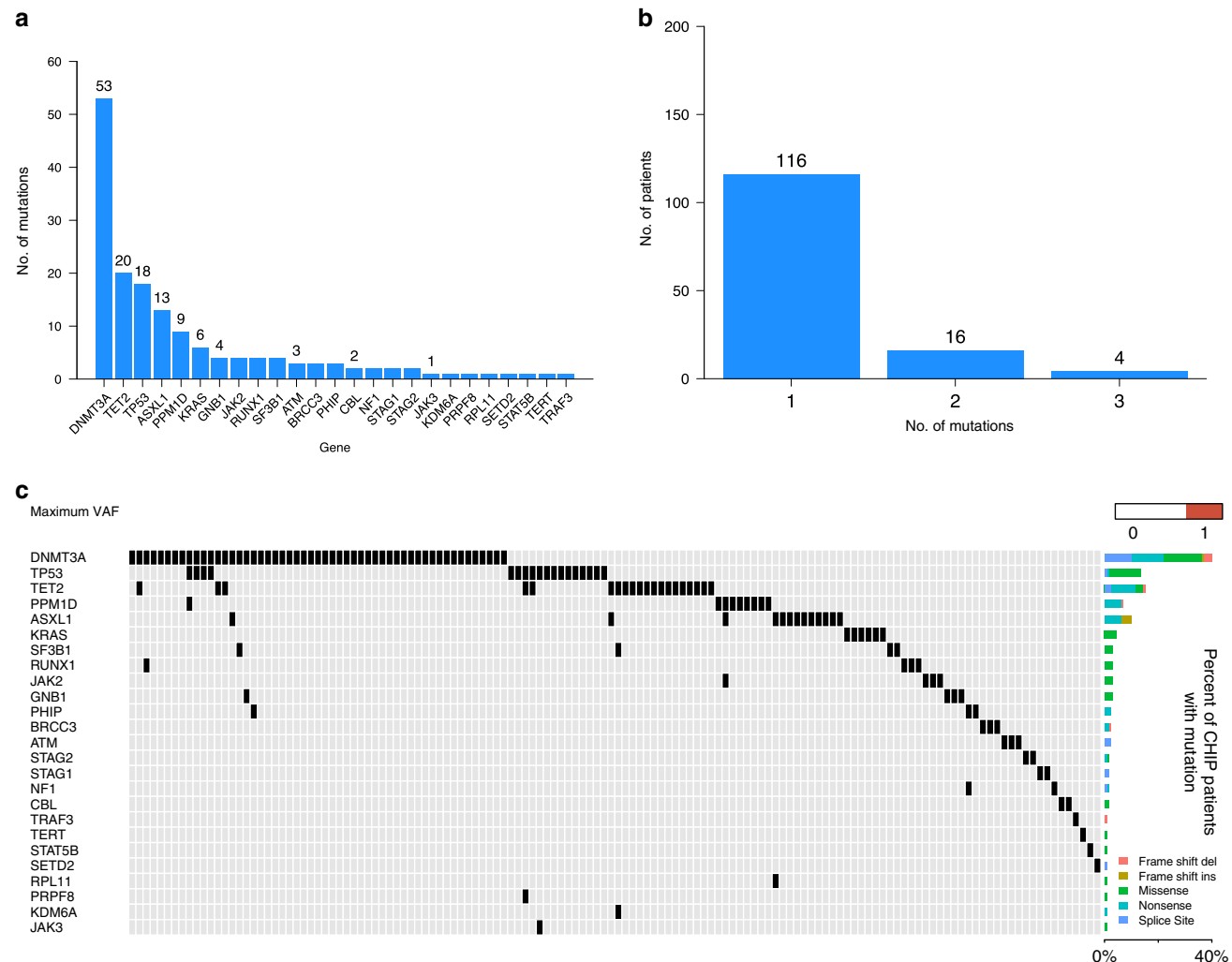

**Fig. 1 Mutational spectrum of CHIP in hematopoietic cells of multiple myeloma patients at the time of autologous stem cell transplant. a** The total number of patients harboring one or more mutations in each gene. **b** Number of patients harboring mutations in 1, 2, and 3 different genes. **c** Comutation plot showing mutations present in all 136 patients: each column represents a single patient. The top row denotes the maximum VAF in each patient, with darker shades of red indicating higher VAF. The VAF cutoff used to call mutations was 0.01. The bar graph on the right designates the percentage of the different mutation subtypes for each gene out of all detected CHIP mutations.

CHIP prior to ASCT was not associated with an increased risk of TMN ($p = 0.4$). However, IMiD maintenance was significantly associated with developing a subsequent TMN ($p = 0.047$) (Fig. 2a, b) and the presence of CHIP did not increase the risk among those receiving first-line IMiD maintenance or IMiD maintenance at any point post-ASCT (Fig. 2c, d). Nine of the 21 patients who developed MDS/AML were still alive at time of analysis. Seven patients had a myeloma relapse before the diagnosis of MDS/AML. Of the 12 patients who died, 5 passed away from the TMN while 7 died from a combination of myeloma progression and MDS/AML.

We next asked whether CHIP at the time of ASCT was clonally related to a subsequent MDS or AML. Sequential samples from 14 of the 21 TMN patients were available for targeted sequencing (Supplementary Tables 5 and 6). 10/14 patients had an identifiable driver mutation at the time of TMN diagnosis, most commonly in *TP53*, but only 4 of those patients had a detectable mutation at the time of ASCT. Among the 6 patients in whom we did not initially find CHIP, we were able to identify the TMN driver mutation in 4 of the ASCT samples at VAFs below our threshold of 0.01. Thus, in 8 of 10 patients with identifiable mutations at the time of TMN diagnosis, at least one somatic

mutation could also be detected prior to ASCT. In most cases, the driver mutation present prior to ASCT had expanded at the time of TMN diagnosis (Supplementary Table 6). It is possible that with deeper sequencing pre-existing mutations could be found in these additional 2 TMN cases as well.

**CHIP is associated with adverse outcomes.** Having found no evidence for higher TMN risk in patients with CHIP, we examined whether CHIP was associated with other adverse outcomes. Out of 629 patients in our cohort, 376 patients had died at the time of analysis. The median OS and progression-free survival (PFS) of our cohort were 7.1 and 2.5 years, respectively. After stratification based on age, International Staging System (ISS) and number of treatment lines prior to ASCT, we modeled the association of CHIP with OS and PFS. The median OS of patients with CHIP was 5.3 years, significantly lower than in those without CHIP (7.5 years) (HR: 1.34, $p = 0.02$, stratified multivariable cox regression model) (Fig. 3a). Interestingly, CHIP was also associated with a lower median PFS of 2.2 years compared to 2.6 years in those without CHIP (HR: 1.45, $p < 0.001$, stratified multivariable cox regression model) (Fig. 3b). We also observed a

**Table 1 Patient characteristics.**

| Characteristics | | CHIP | | |
| --- | --- | --- | --- | --- |
| | **Total**<br>**n = 629 (%)** | **No**<br>**n = 493 (78)** | **Yes**<br>**n = 136 (22)** | **p-value** |
| Age at transplant | | | | |
| Median (range) | 58 (24–83) | 57 (24–72) | 61 (34–83) | <0.001† |
| 20–29 | 2 (0) | 2 (0) | — | <0.001§ |
| 30–39 | 16 (3) | 14 (3) | 2 (1) | |
| 40–49 | 89 (14) | 82 (17) | 7 (5) | |
| 50–59 | 251 (40) | 203 (41) | 48 (35) | |
| 60–69 | 252 (40) | 182 (37) | 70 (51) | |
| 70–79 | 19 (3) | 10 (2) | 9 (7) | |
| Sex | | | | |
| Female | 268 (43) | 219 (44) | 49 (36) | 0.096‡ |
| Male | 361 (57) | 274 (56) | 87 (64) | |
| Race | | | | |
| Asian | 3 (0) | 3 (1) | — | 0.97‡ |
| Black | 21 (3) | 16 (3) | 5 (4) | |
| Hispanic/Latino | 10 (2) | 8 (2) | 2 (1) | |
| Native American | 2 (0) | 2 (0) | — | |
| Other | 6 (1) | 4 (1) | 2 (1) | |
| Unknown | 11 (2) | 9 (2) | 2 (1) | |
| White | 576 (92) | 451 (91) | 125 (92) | |
| Myeloma subtype | | | | |
| Biclonal | 5 (1) | 3 (1) | 2 (1) | 0.42‡ |
| IgA Kappa | 69 (11) | 55 (11) | 14 (10) | |
| IgA Lambda | 50 (8) | 41 (8) | 9 (7) | |
| IgA-only | 7 (1) | 6 (1) | 1 (1) | |
| IgG Kappa | 247 (39) | 199 (40) | 48 (35) | |
| IgG Lambda | 93 (15) | 67 (14) | 26 (19) | |
| IgG-only | 15 (2) | 12 (2) | 3 (2) | |
| Kappa-only | 73 (12) | 61 (12) | 12 (9) | |
| Lambda-only | 55 (9) | 38 (8) | 17 (12) | |
| Non-secretory | 15 (2) | 11 (2) | 4 (3) | |
| 0–25 | 141 (22) | 114 (23) | 27 (20) | 0.81§ |
| 26–50 | 158 (25) | 120 (24) | 38 (28) | |
| 51–75 | 142 (23) | 108 (22) | 34 (25) | |
| 76–100 | 158 (25) | 129 (26) | 29 (21) | |
| ISS | | | | |
| 1 | 353 (56) | 282 (57) | 71 (52) | 0.67§ |
| 2 | 170 (27) | 123 (25) | 47 (35) | |
| 3 | 106 (17) | 88 (18) | 18 (13) | |
| Bone marrow involvement<br>post-induction (percent) | | | | |
| 0–25 | 488 (78) | 380 (77) | 108 (79) | 0.18§ |
| 26–50 | 43 (7) | 33 (7) | 10 (7) | |
| 51–75 | 18 (3) | 12 (2) | 6 (4) | |
| 76–100 | 12 (2) | 7 (1) | 5 (4) | |
| Missing | 68 (11) | 61 (12) | 7 (5) | |
| Induction therapy | | | | |
| Bortezomib | 212 (34) | 167 (34) | 45 (33) | 0.90‡ |
| Cyclophosphamide | 73 (12) | 56 (11) | 17 (12) | |
| Lenalidomide/Thalidomide | 304 (48) | 240 (49) | 64 (47) | |
| Other | 40 (6) | 30 (6) | 10 (7) | |
| Number of therapies prior to<br>induction | | | | |
| 0 | 434 (69) | 339 (69) | 95 (70) | >0.99§ |
| 1 | 118 (19) | 97 (20) | 21 (15) | |
| 2 | 44 (7) | 32 (6) | 12 (9) | |
| 3+ | 33 (5) | 25 (5) | 8 (6) | |
| Family history of any cancers | | | | |
| No | 210 (33) | 162 (33) | 48 (35) | 0.61‡ |
| Yes | 419 (67) | 331 (67) | 88 (65) | |
| Smoking | | | | |
| No | 394 (63) | 319 (65) | 75 (55) | 0.043‡ |
| Yes | 228 (36) | 168 (34) | 60 (44) | |
| Missing | 7 (1) | 6 (1) | 1 (1) | |

†Wilcoxon rank-sum test (two-sided), §Kruskal-Wallis trend test (one-sided), ‡Fisher's exact test (two-sided).

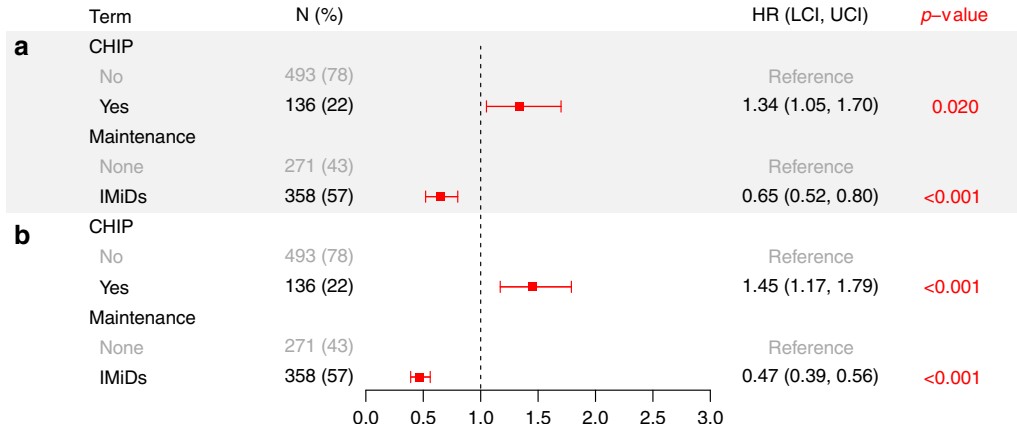

**Fig. 2 Outcome of immunomodulator maintenance and CHIP in the context of therapy related myeloid neoplasms (TMN). a** Cumulative incidence of MDS/AML in patients with IMiD maintenance versus patients without maintenance, with death as an absorbing competing risk. Cumulative incidence of developing MDS/AML among patients with CHIP vs. no CHIP, with death as an absorbing competing risk, **b** among all patients, **c** patients receiving first-line IMiD maintenance and **d** patients receiving IMiDs at any point post ASCT. Groups were tested for equality using a two-sided Gray's test to compare subdistributions for each competing risk.

**Fig. 3 Multivariable cox regression model of OS and PFS. a** OS and **b** PFS models for all 629 patients after stratifying by age, ISS and number of lines of therapy prior to ASCT to investigate the effect of CHIP and IMiD maintenance on outcome. Two-sided Wald p-values are shown for each model coefficient with significant effects displayed in red. Exact *p*-values: A: 0.0197276361 and 0.0000692041; B: 0.0007310355 and 1.547737 × 10$^{-15}$. HR: Hazard Ratio; LCI: Lower Confidence Interval; UCI: Upper Confidence Interval.

worse OS and PFS in patients carrying more than one mutation (Supplementary Fig. 5).

Unlike prior studies of CHIP, we did not observe excess deaths related to cardiovascular disease or stroke, possibly due to the aggressive nature of multiple myeloma[8,19]. The most common cause of death was myeloma disease progression followed by respiratory failure and sepsis. Interestingly, patients with CHIP responded less well to induction therapy such that CHIP was associated with a higher post-induction median level of β2-microglobulin (2.3 mg/dL in those with CHIP compared to 2.0 mg/dL in those without [$p = 0.008$]), and a smaller percentage decrease in M-spike level ($p = 0.008$) post-induction as compared to diagnosis (Supplementary Tables 7–9).

**IMiD maintenance is associated with improved outcomes in all patients.** Treatment with IMiDs has been reported to increase the risk of secondary malignancies including MDS and AML[2–4]. Almost all patients received thalidomide or lenalidomide at some point throughout the course of their disease. Therefore, we asked whether treatment with IMiDs was associated with adverse outcomes in patients with CHIP. Only 57% of the patients in our cohort received first-line IMiD maintenance, with 22% receiving it for at least 3 years (range: 0.1–14.9). First-line IMiD maintenance was associated with a longer median OS of 8.5 years compared to 5.6 years in those not receiving IMiD maintenance [HR: 0.65 (0.52–0.80), $p < 0.001$] (Fig. 3a). As expected, IMiD maintenance was associated with a longer PFS of 3.4 years compared to 1.5 years in those not receiving IMiD maintenance [HR: 0.47 (0.39–0.56), $p < 0.001$] (Fig. 3b). Only 16 patients received proteasome inhibitor-based maintenance, too few to draw significant conclusions (Supplementary Fig. 6).

In patients not receiving IMiD maintenance, CHIP was associated with a significantly lower median OS of 3.6 years compared to 6.6 years in those who did not have CHIP mutations ($p = 0.013$, stratified analysis). However, there was no significant difference in those who received maintenance (median OS of 7.7 and 8.9 years for CHIP and no CHIP, respectively, $p = 0.49$, stratified analysis) (Fig. 4a). Similarly, in patients not receiving IMiD maintenance, CHIP was associated with a significantly lower median PFS of 1.1 years compared to 1.8 years ($p < 0.001$, stratified analysis). There was also no difference in PFS among patients with and without CHIP who received IMiD maintenance (median PFS of 3.3 and 3.6 years for CHIP and no CHIP, respectively, $p = 0.59$, stratified analysis) (Fig. 4b). These results suggest the presence of an interaction between the effect of having

CHIP and IMiD maintenance, which is significant when it comes to PFS outcome [HR: 0.51 (0.34–0.79), $p = 0.002$] (Supplementary Fig. 7).

We next asked whether mutations in specific genes were associated with worse outcomes. The two most commonly mutated genes were *DNMT3A* and *TET2*, which were associated with a significantly reduced PFS and OS as compared to patients without CHIP in the absence of IMiD maintenance (Supplementary Fig. 8). In particular, patients with the p.R882 *DNMT3A* mutation had the worst median OS of 1 year ($p = 0.008$, stratified analysis) and median PFS of 0.9 years ($p = 0.007$, stratified analysis) compared to patients without CHIP (Supplementary Fig. 9). However, in patients who received IMiD maintenance, the decrease in OS and PFS seen in p.R882 patients was completely abrogated.

Altogether, these data suggest that the presence of CHIP at time of ASCT does not increase the risk of TMN associated with IMiD maintenance and that patients with CHIP, when treated with IMiD maintenance, obtain a survival benefit similar to that seen in MM patients generally.

## Discussion

Here we report the first study to date examining the relationship between CHIP and clinical outcomes in MM, involving over 600 patients undergoing ASCT at a single center with a median follow-up of 9.7 years. The frequency of CHIP in this cohort is lower than that seen in patients with relapsed NHL undergoing ASCT and had a mutational spectrum more similar to that seen in healthy adults[7,8,19]. The differences observed between CHIP in MM and NHL are potentially related to the shorter duration of chemotherapy exposure in patients receiving induction therapy for MM and less use of DNA damaging agents that may select for specific mutant clones. The mutations seen most frequently in patients with relapsed NHL are found in *PPM1D* and *TP53*, two genes known to play important roles in the response to DNA damage and chemotherapy resistance[30–32].

TMN is one of the most feared complications of treatment for MM and ASCT, in particular, and the presence of mutant hematopoietic clones has been proposed to herald its development. However, we did not observe an increased risk of TMN in patients with CHIP undergoing ASCT. This may be because the mutations selected for by MM induction therapy do not carry a high risk of subsequent myeloid malignancy[14,15]. Consistent with this hypothesis, the most common mutations in those who developed a TMN were in *TP53*, which was found relatively

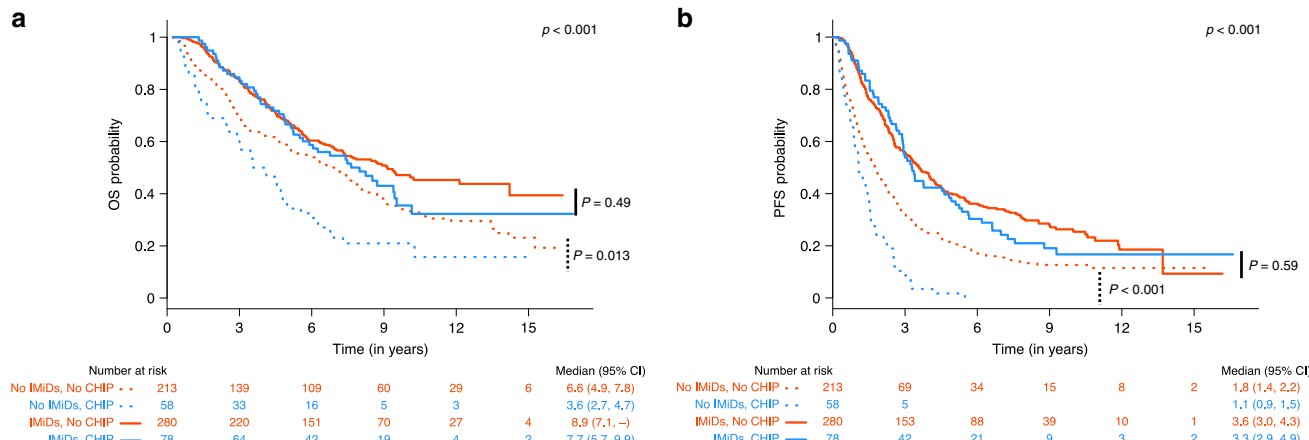

**Fig. 4 Overall survival and progression free survival of patients with respect to IMiD maintenance and CHIP. a** OS and **b** PFS among patients with CHIP versus those without CHIP in the context of receiving versus not receiving IMiD maintenance post ASCT. Overall and pairwise two-sided log-rank *p*-values are shown unadjusted for multiple testing.

infrequently (2.9%) in our cohort at time of ASCT. Further prospective studies are needed to determine whether the presence of *TP53* or other specific mutations at the time of ASCT carry a high risk of TMN development and thus might warrant avoidance of ASCT in MM patients harboring them.

We detected CHIP in 21.6% of MM patients at the time of ASCT and found it to be associated with both decreased OS and PFS. In contrast to previous reports, this was not due to an increased risk of TMN or cardiovascular disease[19]. Unlike in NHL where ASCT is a potentially curative therapy and many patients may expect a long-life expectancy after transplant, MM almost uniformly relapses and may provide insufficient time for cardiovascular disease to manifest. As treatment for MM continues to improve with a concomitant increase in patient life expectancy the potential long-term non-hematopoietic risks associated with CHIP will need to be evaluated further.

Surprisingly, the primary effect of CHIP on survival was due to an increased risk for myeloma progression. Consistent with this finding, patients with myeloma and CHIP had a higher level of $\beta_2$-microglobulin and a smaller percentage decrease in their M-protein post induction, compared to those without CHIP. There are several potential mechanisms by which CHIP could promote increased progression of MM. It is possible that patients with CHIP are more prone to the development of cytopenias and other toxicity from therapy, thus increasing the frequency of treatment delays or dose reductions and limiting their ability to receive optimal myeloma directed therapy. Alternatively, the presence of CHIP could alter the bone marrow (BM) microenvironment in such a way as to promote MM progression. Myeloid cells carrying mutations in *TET2* and *DNMT3A* have been reported to stimulate inflammation through upregulation of IL-1β and IL-6[11,33]. Whether a hyperinflammatory phenotype within the BM niche might favor the growth of MM cells and promote more aggressive disease is a prospect that will require further investigation in both animal models and patients.

IMiD maintenance has demonstrated a clear survival advantage in the post-ASCT setting[2–4]. However, it has also been associated with an increased risk of TMN and mutations in *TP53* have been reported to promote clonal expansion and the development of resistance in del(5q) MDS treated with lenalidomide[34,35]. While induction with lenalidomide could have led to the relative enrichment of TP53 at time of ASCT, we surprisingly saw no increase in TMNs in patients with CHIP and the use of IMiD maintenance. In fact, IMiD maintenance was not only associated with an improvement in both PFS and OS but it completely abrogated the deleterious effects of CHIP in the post-ASCT setting. The findings from this study are limited by the fact that it is retrospective and includes patients treated during the introduction of IMiD maintenance and thus not all patients received maintenance therapy. Follow-up studies examining the large randomized studies of placebo vs. lenalidomide maintenance could further define the role of IMiDs in the survival of patients with CHIP. In addition, because all patients received ASCT, the role of high dose melphalan and stem cell transplant cannot be fully dissected in this study and warrants further investigation within trials comparing clinical outcomes of upfront vs. delayed/salvage ASCT or transplant-based vs. drug-based consolidation[36–40].

In summary, we found CHIP to be a common entity among MM patients undergoing ASCT. The presence of CHIP was associated with worse outcomes and thus, it would be tempting to screen newly diagnosed MM patients for CHIP before ASCT. However, our data suggest that ASCT, when followed by IMiD maintenance, can be safely utilized regardless of CHIP status. Further well-controlled prospective clinical trials are needed to investigate the interaction between CHIP, transplant and IMiDs on outcomes in multiple myeloma.

## Methods

**Cohort**. Following institutional review board (IRB) approval, we collected the clinical data and all available cryopreserved products of mobilized autologous stem-cells from 629 MM patients who underwent ASCT between January 2003 and December 2011 at the Dana-Farber Cancer Institute (DFCI) in Boston, MA. The cutoff date of 2011 was used to enable enough years of follow up following stem cell transplantation and allow for the monitoring of TMNs and survival data. Clinical information was collected through November 2019. The study design complied with the Declaration of Helsinki and International Conference on Harmonization Guidelines for Good Clinical Practice. All subjects previously provided written informed consent to allow the collection of clinical information and genetic analysis of PB and BM samples for research purposes (DF/HCC IRB 01–206, 07–150 and 16–529).

While all 629 patients received ASCT, 21 of those patients received tandem ASCT, and three received a second ASCT at a later time point. Also, 38 patients received allogenic stem cell transplant, seven of which were tandem, and the rest got them at a later time point post relapse.

**Genomic studies**. Deep targeted sequencing was performed on the stem-cell products of 629 MM patients, as well as on available samples of PB and BM aspirates obtained from 15 patients at the time of pre-mobilization and when they developed a hematologic second primary malignancy post-ASCT, respectively. A custom target bait panel of 224 genes was used, including, pan-cancer, myeloma and myeloid malignancy-associated genes (Agilent SureSelectXT hybrid capture system). See Supplementary Table 1 for a list of genes and coordinates. Libraries for the stem cell products were constructed automatically, using the Agilent Bravo robot, and were sequenced on the Illumina HiSeq 4000 platform in pools of 32 samples, achieving a 978X total depth of coverage. Libraries of PB and BM samples were constructed manually and were sequenced on the Illumina HiSeq 2500 platform in pools of 24, achieving 556X total depth of coverage. To detect large-scale copy number alterations that reflect tumor cells within the stem cell products, we also performed ULP-WGS at an average genome-wide fold coverage of 0.1×. Detailed information on library preparation, sequencing platforms and computational analysis for targeted sequencing and ULP-WGS is provided in the supplementary information.

Sequencing was done at the Broad Institute, Cambridge, MA, USA.

**Computational analyses**. Sequencing data was analyzed using the pipelines of the Broad Institute of Harvard and MIT (Firehose, www.broadinstitute.org/cancer/cga). To estimate the presence of tumor, we performed ultra-low pass whole-genome sequencing (ULP-WGS) of all samples to an average genome-wide fold coverage of 0.1×[41]. The depth of coverage was determined using ichorCNA[41–43], to estimate large-scale copy number alterations (CNAs) and the fraction of tumor in ULP-WGS. Low coverage samples (<0.05×) were manually reviewed to determine tumor fraction. All samples had a low tumor fraction (3–5.4%).

The targeted sequencing data of our samples were aligned using BWA-mem and the base qualities of the aligned data were re-calibrated using GATK3 Base Quality Score Recalibration (BQSR)[44,45]. We have utilized the Getz Lab CGA WES Characterization pipeline (https://github.com/broadinstitute/CGA_Production_Analysis_Pipeline) developed at the Broad Institute to call, filter and annotate somatic mutations and copy number variation. We modified this pipeline to call blood samples without matched controls. Hence, we employed the following tools: MuTect[46], Strelka[47], Orientation Bias Filter[48], MAFPonFilter[49], RealignmentFilter, ABSOLUTE[50], GATK MuTect2[51], PicardTools[51,52], Variant Effect Predictor[53], and Oncotator[54].

Usually the variant allele frequency (VAF) cutoff to call mutations is set at 0.02, below which it would be difficult to distinguish somatic mutations from contamination by other samples and sequencing artifacts. Thus, we aimed at estimating contamination to assure that the mutations are biologically relevant even when going below 0.02. We present a framework that allows one to include smaller clones in CHIP calling as a function of the single-sample contamination, providing greater resolution into the initial claim that CHIP is the presence of characteristic driver gene mutations in hematopoietic cells that occurs at a VAF of at least 0.02. This distinction of clonal mutations from contamination becomes especially important for those samples whose sample-to-sample contamination is in fact greater than 2% such that a mutation with a VAF of 0.02 would not be considered in our analysis if the sample's contamination was about 3% (the range of contaminations in our samples was from roughly 1–8%). However, given our high depth of coverage (978×) we were able to confidently call mutations, distinct from sample-to-sample contamination and sequencing artifacts.

To estimate contamination of single samples, we used VerifyBamID[55] using the ExAC[56] VCF to test for germline SNPs with a minimum allele fraction of 0.25. In order to control for noise/artifacts with indel calling, we selected the youngest sample with no detectable CHIP mutations to use as an unmatched control for Strelka and MuTect2. Variants were classified as pathogenic driver mutations based on mutation type, position, and frequency in published reports[8,19,57] and public

databases[58]. The set of rules to consider a queried mutation as a "driver" mutation is outlined in supplementary table 2. The minimum number of alternate reads we chose to accept or reject to call a variant according to MuTect2 was not a hard cutoff but rather one determined as a function of the read depth, strand biases and contamination at a given site[46]. Consequently, the lowest number of accepted alternate reads was 4. Variants with allele fraction less than 0.01 were excluded and, except for *DNMT3A, TET2, ASXL1, PPM1D, TP53, JAK2, SF3B1, SRSF2*, mutations with VAF above 0.35 were also excluded since these often represent germline polymorphisms. To further confirm that our called mutations are somatic drivers, we excluded single nucleotide variant (SNV) mutations with a TLOD score below 6.3 (via MuTect2) and Insertion-Deletion (Indel) mutations with a QSI_NT score below 30 (via Strelka). As a consequence of bait preferences, some germline mutations can sometimes have putatively somatic allele fractions if they are in regions of poor mapping. However, because Strelka and MuTect2 perform local realignment, these callers do not have deflated allele fractions in contrast to other callers that double count reference reads if they detect split reads, which is a consequence of structural variation. Finally, all variants were visually inspected in Integrated Genome Viewer (IGV)[59].

We compared sequential samples taken from the same patient, in search for the presence of a mutation, called in one sample, in another sample from the same patient. To do that, we performed force-calling which is a technique that looks at the reads in a BAM file from a list of genomic coordinates and calculates the number of reads supporting an alternate allele at that location[60].

**Statistical analyses**. OS was defined as the time from transplantation until death from any cause, with censoring at time last known to be alive. PFS was measured from ASCT to the date of disease progression or death from any cause, censoring at time last known to be alive and progression-free. Survival curves were estimated using the Kaplan Meier method, with variance and CIs estimated using Greenwood's formula. Stratified Cox regression was used for time-to-event outcomes; hazard ratios (HR) and 95% confidence intervals (CI) were reported. Stratification was based on age (age groups: 20–29, 30–39, 40–49, 50–59, 60–69, 70–79), ISS of the disease[61] and number of lines received prior to ASCT. *p*-Values were two-sided, and those with <0.05 were considered statistically significant. All data were analyzed using R version 3.5.0 (R Core Team).

Death and occurrence of a TMN (namely, MDS and AML) were modeled as competing events. Wilcoxon rank-sum and Fisher's exact tests were used for CHIP association with continuous and categorical variables, respectively. Ordinal variables with three or more groups were tested for association with CHIP using a Kruskal-Wallis test for singly-ordered contingency tables.

**Reporting Summary**. Further information on research design is available in the Nature Research Reporting Summary linked to this article.

## Data availability

Sequencing data and basic phenotype data can be accessed via dbGaP accession code phs001323. All other remaining data are available within the article and supplementary files, or available from the authors upon request.

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

## Acknowledgements

We would like to thank Darlys Schott and Doreen Hearsey, as well as the Connell O'Reilly Cell Manipulation Facility and the Ted and Eileen Pasquarello Tissue Bank, for assistance in obtaining patient samples. We would also like to thank the International Myeloma Society (IMS) for their support. This work was supported by grants from the Multiple Myeloma Research Foundation (MMRF), Adelson Medical Research Foundation (AMRF), Stand Up to Cancer (SU2C) and the Leukemia and Lymphoma Society (LLS) awarded to Dr. Irene M. Ghobrial.

## Author contributions

Conceptualization, T.H.M., A.S.S., C.J.G., S. Manier, B.L.E., D.S.N., and I.M.G.; Methodology, J.P., R.R., M.L., R.S.P., C.L., S.H.A., B.R., E.M.V.A., J.J.K., C.S., S. Mehr, L.T., G.G., and D.S.N.; Investigation, T.H.M., A.H.N., M.C., D.H., M.B., S.T., K.H., H.D., M.M.I., and C.J.B.; Writing – Original Draft, T.H.M.; Writing – Review & Editing, T.H.M., A.S.S., C.J.G., D.A., R.L.S., N.M., K.C.A., D.P.S., J.P.L., P.G.R., J.R., B.L.E., R.J.S., L.T., D.S.N., and I.M.G.; Funding acquisition, I.M.G.; Resources, G.G., D.S.N., and I.M.G.; Supervision, D.S.N. and I.M.G.

## Competing interests

The authors declare the following competing interests: M.B.: Dava Oncology: Honoraria. N.C.M.: OncoPep: Other: Board of director. K.C.A.: Gilead: Membership on an entity's Board of Directors or advisory committees; OncoPep: Equity Ownership, Scientific founder; Celgene: Consultancy; C4 Therapeutics: Equity Ownership, Scientific founder; Bristol Myers Squibb: Consultancy; Millennium Takeda: Consultancy. P.G.R.: Oncopeptides: Membership on an entity's Board of Directors or advisory committees; BMS: Research Funding; Janssen: Membership on an entity's Board of Directors or advisory committees; Karyopharm: Membership on an entity's Board of Directors or advisory committees; Jazz Pharmaceuticals: Membership on an entity's Board of Directors or advisory committees; Celgene: Membership on an entity's Board of Directors or advisory committees, Research Funding; Celgene: Membership on an entity's Board of Directors or advisory committees, Research Funding; Amgen: Membership on an entity's Board of Directors or advisory committees; Takeda: Membership on an entity's Board of Directors or advisory committees, Research Funding. Amgen: Research Funding; Equillium: Research Funding; Kite/Gilead: Research Funding; Aleta Biotherapeutics: Consultancy; Avrobio: Consultancy; Celgene: Consultancy; Falcon Therapeutics: Consultancy; Talaris Therapeutics: Consultancy; TScan Therapuetics: Consultancy. B.L.E.: Celgene: Research funding; Deerfield: Research funding; GRAIL: Consultancy; Exo Therapeutics: scientific advisory boards; Skyhawk Therapeutics: scientific advisory boards. R.J.S.: Kiadis: Board of Directors; Juno: DSMB; Gilead: Consultancy; Neovii: Consultancy; Cugene: Consultancy; Jazz: Consultancy; Mana Therapeutics: Consultancy; VOR: Consultancy; Novartis: Consultancy. G.G.: receives research funds from IBM and Pharmacyclics and is an inventor on patent applications related to MuTect, ABSOLUTE, MutSig and POLYSOLVER. I.M.G.: Celgene: Consultancy; Janssen: Consultancy; BMS: Consultancy; Takeda: Consultancy; GSK: Consultancy; Sanofi: Consultancy; Karyopharm: Consultancy; Abbvie: Consultancy; Cellectar: Consultancy; Genentech: Consultancy; Adaptive: Consultancy. The remaining authors declare no competing interests.
