## [Peer Review File · Nature Communications]

Reviewers' comments:

Reviewer #1 (Remarks to the Author):

This is an important and interesting paper. The authors conclude that IMiDs can be safely administered to myeloma patients with CHIP. Moreover, the study reveals CH to be a prognostic factor in myeloma and implies the ability to intervene upon or modulate/cancel the adverse effects of CHIP via IMiDs. Some concerns remain with regard to publishing the manuscript in its current form.

MAJOR ITEMS:

- The prevalence of CHIP in the introduction should be refined. The seminal Jaiswal/Genovese work used lower depth sequencing and was underpowered to detect small CHIP clones. The true prevalence is higher with deep sequencing. Please update this number to reflect a more updated estimate derived from the more recent literature using deep sequencing such as from the labs of Drs. Levine, Druley, Hassane, and Shlush. In general, more clarity around true differences in prevalence vs. differences in power would improve the quality and authority of the manuscript.
- The discussion states that the CHIP prevalence in MM is different than DLBCL (lower). This statement requires an evaluation of sensitivity/methods between the two studies and should be constrained to genes/regions/exons commonly interrogated by the studies (e.g. DTA mutations plus TP53, PPM1D).
- Figure 1C should show the percentage of patients affected with each mutation and clearly indicate the VAF cutoff used for the figure.
- The authors state "mutational spectrum is similar to that seen in healthy individuals in the general population". However, the prevalence of TP53-driven CH seems high compared to previous work relative to TET2. TP53 shows nearly the same prevalence as TET2 in the current manuscript. In the Jaiswal/Genovese papers, TP53 mutations are very rare compared to TET2. In Coombs et al, the prevalence of TP53 driven CHIP was approximately 1/3rd that of TET2. In the Abelson/Desai papers, healthy controls not proceeding to AML are practically devoid of TP53 mutations in peripheral blood. On the other hand, TP53 can be somatically mutated in myeloma and can be antecedent to MDS/AML. The authors should include an analysis/discussion of this difference and provide an explanation.
- In figure S1B, there is a secondary peak for frameshift variants at about 30% (unless it just appears this way due to the parameters selected for kernel density estimation in the figure). This is unusual. Depending on the size of the alteration, frameshift mutations can demonstrate a lower apparent VAF than actual VAF due to lower capture efficiency for the mutant allele with the capture probes (especially in the absence of bait tiling). Thus, this second peak may be truly germline. How is the possibility excluded? The authors should determine the allelic fraction for some these cases using an orthogonal method (e.g. ddPCR) or otherwise demonstrate that the capture efficiency bias noted as a possibility here does not exist.
- What genes drive the association between CHIP and smoking? Is ASXL1 seen more often in smokers?
- IMiD therapy is standard of care for eligible MM patients. From where is the IMiD-free arm derived? Are the IMiD-free patients less fit and thus ineligible? Does this contribute to their worse outcomes with CHIP?
- Expanding on the above point, for figure 2, were the patients who received IMiDs receiving more modern therapy compared to patients who received no IMiDs? For the comparison to be valid, all

factors aside from IMiD maintenance need to be balanced in the IMiD-positive and IMiD-negative arm. Please address by stating the balance and/or via the application of multivariable statistics.

- Figure 3: Based on Figure 2 and subsequent Figure 4, one might expect an interaction between IMiD arm and CH. For Figure 3, what is the effect/direction/significance of the interaction? Is it consistent with Figure 2 and 4?

- The overall observation that the adverse impact of CH is moderated by IMiDs is among the most interesting points of this work. CH exhibits a hyperinflammatory phenotype as the authors note in the discussion. IMiDs can demonstrate anti-inflammatory properties (e.g. Quach et al Leukemia 2014). Do IMiDs act via normalization of the inflammatory burden in MM patients? To close the loop, the authors should demonstrate the impact of IMiDs on inflammatory cytokines in the cohort - especially those associated with CHIP (IL-1beta and/or IL-6 depending on the CHIP mutation).

MINOR ITEMS:

- The presentation of this study would benefit from a clear diagram.

- To improve clarity/readability, please state how low pass WGS was used to detect circulating myeloma? What was being sought? The rationale is not intuitive to the average non-bioinformatician/genomics reader. A phrase or sentence can help clarify.

- Please move/copy information about depth of coverage to the main methods in a sentence. Please specify whether this is total depth or unique molecular depth. If it is total depth, please state the unique molecular depth. Technical details regarding variant calling should be clearly stated (number of supporting unique reads, method of artifact filtration, etc).

- The authors should note that it is formally possible that the 2/10 evaluable TMN cases may have had a related CHIP clone with deeper sequencing.

Reviewer #2 (Remarks to the Author):

Predicting outcome of cancer therapy in terms of efficacy and toxicity is an important goal for improving cancer treatment, but it is not currently clear how best to do this. Both ASCT and IMiD maintenance have been shown to be beneficial overall in myeloma, but both carry significant risks of secondary cancers. Work over the past few years has revealed that CHIP may be a predictor of various diseases and, in particular, haematological neoplasms such as MDS and AML. In this manuscript, Mouhieddine and colleagues seek to understand whether the presence of CHIP is a predictor for negative outcome for transplant-eligible myeloma patients treated with ASCT and IMiD maintenance.

Key findings are:

- CHIP at a VAF of 0.01 or above in 21.6% of stem cell harvests. This is very unlikely to be contributed to by myeloma cell contamination. The mutational spectrum appears to be typical for CHIP reported elsewhere.
- Myeloid neoplasms only occurred in the setting of IMiD maintenance.
- Whether or not patients received IMiD maintenance, the presence of CHIP did not affect the risk of developing MDS/AML.
- 21 patients developed MDS/AML. For the 10 patients with a TMN driver, this driver was found at time of ASCT in 8. 4/6 patients without defined CHIP had a driver at ASCT when the VAF threshold

was lowered.

- In multivariate analysis, CHIP was associated with shorter OS and PFS, which appeared to be predominantly due to myeloma progression, whereas IMiD maintenance was associated with longer OS and PFS.
- It appears that the use of IMiD maintenance overcomes the negative effects of CHIP on outcome, either in general or in the presence of the commonest mutations, DNMT3A and TET2.

The major conclusion from this study would appear to be that, whilst CHIP is a predictor for negative outcome of ASCT in myeloma, IMiD maintenance is beneficial regardless of whether the patient has CHIP. The study has been carried out to a very high standard and is clearly presented. It provides novel understanding of the role of CHIP in myeloma, has important clinical implications (although the authors are careful to point out that this is a retrospective study), and further insights into treatment stratification in cancer generally.

Minor comment

My only comment would be that CHIP is essentially defined by a VAF cut-off, which is a function of sequencing depth (e.g. the authors were able to use a threshold of 0.01, rather than 0.02, because they had sequenced in higher depth to previous studies). Interestingly, when the VAF threshold was lowered, 4/6 TNM patients who did not have CHIP were found to have the driver mutation at the time of ASCT. These patients essentially had CHIP, but at a level below the threshold of detection. I think for the benefit of some non-specialist readers of Nature Communications, it may be worth a couple of sentences in the Discussion to highlight these issues and to consider how greater sequencing depth and a lower VAF threshold may have impacted results (particularly the AML/MDS association). However, this is a relatively minor point and I would understand if this discussion were precluded by the word count.

REBUTTAL LETTER

Reviewers' comments:

Reviewer #1 (Remarks to the Author):

This is an important and interesting paper. The authors conclude that IMiDs can be safely administered to myeloma patients with CHIP. Moreover, the study reveals CH to be a prognostic factor in myeloma and implies the ability to intervene upon or modulate/cancel the adverse effects of CHIP via IMiDs. Some concerns remain with regard to publishing the manuscript in its current form.

MAJOR ITEMS:

- The prevalence of CHIP in the introduction should be refined. The seminal Jaiswal/Genovese work used lower depth sequencing and was underpowered to detect small CHIP clones. The true prevalence is higher with deep sequencing. Please update this number to reflect a more updated estimate derived from the more recent literature using deep sequencing such as from the labs of Drs. Levine, Druley, Hassane, and Shlush. In general, more clarity around true differences in prevalence vs. differences in power would improve the quality and authority of the manuscript.

We agree with the reviewer's comment and thank them for pointing out this important issue. There have been a number of studies of clonal hematopoiesis using a variety of differing technologies including whole genome sequencing, whole exome sequencing and targeted panels. Direct comparisons between studies have been made difficult by these differences in technical approaches and the genes analyzed in each study. Each of these studies have used differing definitions of clonal hematopoiesis, that is dependent on the sequencing depth, the set of genes sequenced, and the VAF cutoff. We have added further discussion of these issues to the introduction (page 3, paragraph 3):

In these initial large studies, the prevalence of CHIP was found to increase with age, with a prevalence of approximately 1% in individuals under the age of 50 but 10-15% in those older than 70¹⁻³. Because these studies used whole genome or whole exome sequencing, they had a limited ability to identify small clones with a variant allele fraction (VAF) less than 0.02. More recent studies utilizing molecular barcodes and deep targeted sequencing have identified somatic mutations at VAFs as low as 0.0001, demonstrating that small clones can be detected in virtually all adults⁴. However, the clinical significance of these small clones is unclear as the presence of mutations at a VAF greater than 0.01 carried the strongest association with the development of a subsequent myeloid malignancy⁴⁻⁸. Additional work is ongoing to identify clinically meaningful VAF cutoffs, which also may vary depending upon the specific clinical scenario and the patient outcomes being measured.

References:

1. Zink F, Stacey SN, Norddahl GL, et al. Clonal hematopoiesis, with and without candidate driver mutations, is common in the elderly. *Blood* 2017;130:742-52.
2. Genovese G, Kahler AK, Handsaker RE, et al. Clonal hematopoiesis and blood-cancer risk inferred from blood DNA sequence. *The New England journal of medicine* 2014;371:2477-87.
3. Jaiswal S, Fontanillas P, Flannick J, et al. Age-related clonal hematopoiesis associated with adverse outcomes. *The New England journal of medicine* 2014;371:2488-98.
4. Young AL, Challen GA, Birmann BM, Druley TE. Clonal haematopoiesis harbouring AML-associated mutations is ubiquitous in healthy adults. *Nature communications* 2016;7:12484.
5. Abelson S, Collord G, Ng SWK, et al. Prediction of acute myeloid leukaemia risk in healthy individuals. *Nature* 2018;559:400-4.
6. Desai P, Mencia-Trinchant N, Savenkov O, et al. Somatic mutations precede acute myeloid leukemia years before diagnosis. *Nature medicine* 2018;24:1015-23.
7. Young AL, Tong RS, Birmann BM, Druley TE. Clonal hematopoiesis and risk of acute myeloid leukemia. *Haematologica* 2019;104:2410-7.
8. Wong TN, Ramsingh G, Young AL, et al. Role of TP53 mutations in the origin and evolution of therapy-related acute myeloid leukaemia. *Nature* 2015;518:552-5.

- The discussion states that the CHIP prevalence in MM is different than DLBCL (lower). This statement requires an evaluation of sensitivity/methods between the two studies and should be constrained to genes/regions/exons commonly interrogated by the studies (e.g. DTA mutations plus TP53, PPM1D).

We agree with the reviewer's comment that direct comparisons between studies is challenging given differences in sequencing technology and sensitivity. However, one of the reasons we compared our results to that of the NHL study is not only because it is an analogous clinical setting (pre-auto transplant), but because we utilized similar sequencing methods and analytic pipelines. In addition, the gene sets are largely overlapping, only with the addition of a small number of extra genes in our panel which led to the identification of only 2 extra patients with CHIP, carrying mutations that would not have been identified by the NHL study. The major difference between the two studies is the VAF cutoff which was 0.02 for NHL and 0.01 for MM. However, when we use the same VAF cutoff of 0.02, CHIP prevalence in MM is 14% (compared to 29% in the NHL study) as already stated in our results section. Thus, despite any differences in sequencing approaches we believe that CHIP prevalence in MM prior to ASCT is lower than that of NHL.

- Figure 1C should show the percentage of patients affected with each mutation and clearly indicate the VAF cutoff used for the figure.

The numbers of patients harboring each mutation was indicated in figure 1A as well as on the right side of figure 1C, where the bar graphs indicate the number of patients affected by each mutation. Thus, we will change the X-axis of those bars to percentage in order to reflect the percentage of patients affected by each mutation out of all CHIP patients. Figure 1C has been modified to the below figure. We have also added the phrase "The VAF cutoff used to call mutations was 0.01" to the figure legend of figure 1.

- The authors state "mutational spectrum is similar to that seen in healthy individuals in the general population". However, the prevalence of TP53-driven CH seems high compared to previous work relative to TET2. TP53 shows nearly the same prevalence as TET2 in the current manuscript. In the Jaiswal/Genovese papers, TP53 mutations are very rare compared to TET2. In Coombs et al, the prevalence of TP53 driven CHIP was approximately 1/3rd that of TET2. In the Abelson/Desai papers, healthy controls not proceeding to AML are practically devoid of TP53 mutations in peripheral blood. On the other hand, TP53 can be somatically mutated in myeloma and can be antecedent to MDS/AML. The authors should include an analysis/discussion of this difference and provide an explanation.

We thank the reviewer for pointing out this issue. Given the caveats associated with comparing between studies, there is still likely to be an increased predominance of *TP53* mutations in our cohort as compared to healthy adults. However, the mutational spectrum is clearly more similar to that seen in healthy adults than that seen in NHL patients prior to ASCT in whom *PPM1D* and *TP53* mutations predominate. These differences may be due to the differences in prior exposure to cytotoxic therapies. We have edited the text to clarify this point of comparison (page 6, paragraph 2).

While TP53 can be detected in myeloma cells, it is relatively rare in patients with newly diagnosed multiple myeloma and is more common in patients with relapsed or refractory disease. In addition, the presence of circulating myeloma cells in peripheral blood stem cell products is rare especially as most patients undergoing stem cell harvest have decreased myeloma burden due to prior induction chemotherapy. Among those 18 patients with *TP53* mutations, 4 had a complete response and 9 had a very good partial response after induction, further decreasing the likelihood of circulating contaminating tumors cells. Furthermore, we did perform ultra-low pass whole-genome sequencing (ULP-WGS) in order to detect myeloma contamination in all our samples, all of which were below 0.05. This low level of tumor in the peripheral blood samples would be difficult to detect at a VAF of 0.01 using our sequencing technology. Finally, the sequenced samples in this study were collected following induction therapy, which usually includes lenalidomide, an agent known to allow the growth of TP53 CHIP clones. Thus, some of

the TP53 clones observed at time of ASCT, could have been the result of lenalidomide treatment. As such, we have included the following statement in the discussion on page 12 paragraph 1:

While induction with lenalidomide could have led to the relative enrichment of TP53 at time of ASCT, we surprisingly saw no increase in TMNs in patients with CHIP and the use of IMiD maintenance.

- In figure S1B, there is a secondary peak for frameshift variants at about 30% (unless it just appears this way due to the parameters selected for kernel density estimation in the figure). This is unusual. Depending on the size of the alteration, frameshift mutations can demonstrate a lower apparent VAF than actual VAF due to lower capture efficiency for the mutant allele with the capture probes (especially in the absence of bait tiling). Thus, this second peak may be truly germline. How is the possibility excluded? The authors should determine the allelic fraction for some these cases using an orthogonal method (e.g. ddPCR) or otherwise demonstrate that the capture efficiency bias noted as a possibility here does not exist.

We thank the reviewer for pointing out this important issue. The second peak of frameshift mutations represents 4 patients with mutations in ASXL1 (VAFs of 26% and 27%), a DNMT3A mutation with a VAF of 35% and a TRAF3 mutation with a VAF of 32%. Indeed, as the reviewer correctly points out, the VAFs of frameshift mutations might actually be higher. As a consequence of bait preferences, some germline mutations can have putatively somatic allele fractions just because they are in regions of poor mapping (this holds for indels and SNVs). However, because Strelka and MuTect2 perform local realignment, these callers do not have deflated allele fractions in contrast to something like Varscan or indelocator that double counts reference reads if it detects split reads (a consequence of structural variation). Thus, we are confident that these variants are well-mapped and that our allele fractions are not deflated. Furthermore, frameshift mutations in ASXL1 and DNMT3A are known to be pathogenic and a high VAF would still most likely be reflective of a somatic mutation. This was mentioned in our supplementary appendix on page 3: "Variants with allele fraction less than 0.01 were excluded and, except for DNMT3A, TET2, ASXL1, PPM1D, TP53, JAK2, SF3B1, SRSF2, mutations with VAF above 0.35 were also excluded since these often represent germline polymorphisms."

We have also added the following statement to the supplementary appendix on page 4, paragraph 1, to further express our confidence in the reported VAFs:

As a consequence of bait preferences, some germline mutations can sometimes have putatively somatic allele fractions if they are in regions of poor mapping. However, because Strelka and MuTect2 perform local realignment, these callers do not have deflated allele fractions in contrast to other callers that double count reference reads if they detect split reads, which is a consequence of structural variation.

- What genes drive the association between CHIP and smoking? Is ASXL1 seen more often in smokers?

Unfortunately, when studying each mutation separately, the numbers are too low to detect a significant correlation between a specific mutation and smoking. It was only when looking at all mutations collectively that we detect an association between CHIP as a whole and smoking. The

association between CHIP and smoking has been previously described in other large cohorts (Zink et al. Clonal hematopoiesis, with and without candidate driver mutations, is common in the elderly. *Blood*. 2017).

- IMiD therapy is standard of care for eligible MM patients. From where is the IMiD-free arm derived? Are the IMiD-free patients less fit and thus ineligible? Does this contribute to their worse outcomes with CHIP?

We agree with the reviewer that this is an important issue and a limitation of our retrospective approach. The cohort of patients described in this study included all patients seen for the first time at our institution between 2003 and 2011. This time period overlaps with the introduction of IMiD maintenance and thus many patients did not receive maintenance therapy. Thus, many of the patients who did not receive IMiD maintenance would also have had less access to other new therapies upon relapse. We mention this as one of the limitations in the discussion section (page 11, paragraph 3), and it could confound our observation that patients not receiving IMiDs have worse outcomes, although the advantages of IMiD maintenance are well documented in other studies. We would not expect the rates of CHIP to be different based upon the date of diagnosis, thus our comparisons between those with and without CHIP remain valid, although we cannot rule out the unlikely possibility that it is not IMiD maintenance but some other new subsequent therapy available only to those who received IMiDs that accounts for the improvements in outcome we observe.

- Expanding on the above point, for figure 2, were the patients who received IMiDs receiving more modern therapy compared to patients who received no IMiDs? For the comparison to be valid, all factors aside from IMiD maintenance need to be balanced in the IMiD-positive and IMiD-negative arm. Please address by stating the balance and/or via the application of multivariable statistics.

Given that these patients were seen for the first time between 2003 and 2011, the cohort was mainly divided into IMiDs (thalidomide and lenalidomide) and none. Modern therapy was mainly utilized in patients that were diagnosed closer to 2011 and only after their first relapse. While overall survival could be affected by the use of subsequent more modern therapies, the PFS would not be affected by these factors as modern therapy would have been utilized after at least the first relapse.

- Figure 3: Based on Figure 2 and subsequent Figure 4, one might expect an interaction between IMiD arm and CH. For Figure 3, what is the effect/direction/significance of the interaction? Is it consistent with Figure 2 and 4?

Indeed, we agree that these figures reflect the possible presence of an interaction. We have repeated the multivariable cox regression analysis and tested for the interaction. As shown in the above multivariate cox regression table, there seems to be a significant interaction between the IMiD arm and CHIP when it comes to PFS, but not for OS. However, the fact that the interaction is not significant here does not mean that it is null. Thus, PFS (significant interaction) and OS (non-significant interaction) results are still concordant with figures 2 and 4. We have the above figure (Fig.S7) to the supplementary appendix and added the following statement on page 9, paragraph 3:

These results suggest the presence of an interaction between the effect of having CHIP and IMiD maintenance, which is significant when it comes to PFS outcome [HR: 0.51 (0.34-0.79), p=0.002] (Fig.S7).

- The overall observation that the adverse impact of CH is moderated by IMiDs is among the most interesting points of this work. CH exhibits a hyperinflammatory phenotype as the authors note in the discussion. IMiDs can demonstrate anti-inflammatory properties (e.g. Quach et al Leukemia 2014). Do IMiDs act via normalization of the inflammatory burden in MM patients? To close the loop, the authors should demonstrate the impact of IMiDs on inflammatory cytokines in the cohort - especially those associated with CHIP (IL-1beta and/or IL-6 depending on the CHIP mutation).

We agree with the reviewer that this is an intriguing finding and is a very important area for future investigation. We can speculate on a variety of potential mechanisms for how IMiD therapy might alter the inflammatory microenvironment in the setting of CH. The idea that IL-1b or IL-6 elaborated by mutant CH cells might somehow be altered by treatment with IMiDs is one such mechanism. Unfortunately, we do not have samples available for analysis of cytokine levels pre- and post-IMiD therapy. We hope to identify cohorts of patients in which to further investigate these hypotheses in the future but feel that it is beyond the clinical scope of the work presented here. As such we did have the following statement in our discussion on page 11, paragraph 2: “Alternatively, the presence of CHIP could alter the bone marrow microenvironment in such a way as to promote MM progression. Myeloid cells carrying

mutations in *TET2* and *DNMT3A* have been reported to stimulate inflammation through upregulation of IL-1 β and IL-6^{11,34}. Whether a hyperinflammatory phenotype within the bone marrow niche might favor the growth of MM cells and promote more aggressive disease is a prospect that will require further investigation in both animal models and patients.”

MINOR ITEMS:

- The presentation of this study would benefit from a clear diagram.

We have added the below diagram (Figure S1) representing the study workflow to the supplementary appendix.

- To improve clarity/readability, please state how low pass WGS was used to detect circulating myeloma? What was being sought? The rationale is not intuitive to the average non-bioinformatician/genomics reader. A phrase or sentence can help clarify.

We have updated the text to clarify this and it now reads the following on page 6, paragraph 1:

We next sought to verify that the detected somatic mutations were not coming from circulating MM cells, but bone marrow MM samples were not available for sequencing. However, with the exception of TP53, these genes are not recurrently mutated in MM²⁴⁻²⁸ and prior work has demonstrated that plasma cell contamination of peripheral blood stem product is generally minimal^{29,30}. In order to confirm this, we performed ULP-WGS on the stem cell products to detect large-scale copy number alterations, which reflect tumor cells. Only 49 patients had a measurable MM tumor fraction (3-5.4%) in their stem cell product and removal of these samples from statistical analysis did not alter our results.

- Please move/copy information about depth of coverage to the main methods in a sentence. Please specify whether this is total depth or unique molecular depth. If it is total depth, please state the unique molecular depth. Technical details regarding variant calling should be clearly stated (number of supporting unique reads, method of artifact filtration, etc).

We would like to point out that our sequencing technology did not involve UMIs and thus we reported the total depth of coverage. Furthermore, based on Picard Pipeline's MarkDuplicates, our bam files are marked for duplicates and Mutect takes that into account, so we are reporting the deduplicated coverage.

Information regarding artifact filtration is already mentioned on page 3 of the supplementary appendix. However, we were referring to it as "noise". The word "noise" has now been changed to "artifact". Furthermore, we added the following statement on the number of supporting unique reads in the supplementary appendix page 4, paragraph 1:

The minimum number of alternate reads we chose to accept or reject to call a variant according to Mutect was not a hard cutoff but rather one determined as a function of the read depth, strand biases and contamination at a given site (Cibulskis et al. Sensitive detection of somatic point mutations in impure and heterogeneous cancer samples. *Nature Biotechnology*. 2013).

We also added the total depth of coverage to the main methods section, as suggested by the reviewer, as well as expanded the Genomics main methods section to read:

To identify somatic mutations in the stem-cell products and TMN samples, targeted sequencing of 224 genes recurrently mutated in hematologic malignancies was performed at a 978X and 556X total depth of coverage, respectively. To detect large-scale copy number alterations that reflect tumor cells within the stem cell products, we also performed ultra-low-pass whole-genome sequencing (ULP-WGS) at an average genome-wide fold coverage of 0.1X. Detailed information on library preparation, sequencing platforms and computational analysis for targeted sequencing and ULP-WGS is provided in the supplementary appendix.

- The authors should note that it is formally possible that the 2/10 evaluable TMN cases may have had a related CHIP clone with deeper sequencing.

We thank the reviewer for pointing out this important issue and we have updated the text to reflect this.

Reviewer #2 (Remarks to the Author):

Predicting outcome of cancer therapy in terms of efficacy and toxicity is an important goal for improving cancer treatment, but it is not currently clear how best to do this. Both ASCT and IMiD maintenance have been shown to be beneficial overall in myeloma, but both carry

significant risks of secondary cancers. Work over the past few years has revealed that CHIP may be a predictor of various diseases and, in particular, haematological neoplasms such as MDS and AML. In this manuscript, Mouhieddine and colleagues seek to understand whether the presence of CHIP is a predictor for negative outcome for transplant-eligible myeloma patients treated with ASCT and IMiD maintenance.

Key findings are:

- CHIP at a VAF of 0.01 or above in 21.6% of stem cell harvests. This is very unlikely to be contributed to by myeloma cell contamination. The mutational spectrum appears to be typical for CHIP reported elsewhere.
- Myeloid neoplasms only occurred in the setting of IMiD maintenance.
- Whether or not patients received IMiD maintenance, the presence of CHIP did not affect the risk of developing MDS/AML.
- 21 patients developed MDS/AML. For the 10 patients with a TMN driver, this driver was found at time of ASCT in 8. 4/6 patients without defined CHIP had a driver at ASCT when the VAF threshold was lowered.
- In multivariate analysis, CHIP was associated with shorter OS and PFS, which appeared to be predominantly due to myeloma progression, whereas IMiD maintenance was associated with longer OS and PFS.
- It appears that the use of IMiD maintenance overcomes the negative effects of CHIP on outcome, either in general or in the presence of the commonest mutations, DNMT3A and TET2.

The major conclusion from this study would appear to be that, whilst CHIP is a predictor for negative outcome of ASCT in myeloma, IMiD maintenance is beneficial regardless of whether the patient has CHIP. The study has been carried out to a very high standard and is clearly presented. It provides novel understanding of the role of CHIP in myeloma, has important clinical implications (although the authors are careful to point out that this is a retrospective study), and further insights into treatment stratification in cancer generally.

Minor comment

My only comment would be that CHIP is essentially defined by a VAF cut-off, which is a function of sequencing depth (e.g. the authors were able to use a threshold of 0.01, rather than 0.02, because they had sequenced in higher depth to previous studies). Interestingly, when the VAF threshold was lowered, 4/6 TNM patients who did not have CHIP were found to have the driver mutation at the time of ASCT. These patients essentially had CHIP, but at a level below the threshold of detection. I think for the benefit of some non-specialist readers of Nature Communications, it may be worth a couple of sentences in the Discussion to highlight these issues and to consider how greater sequencing depth and a lower VAF threshold may have impacted results (particularly the AML/MDS association). However, this is a relatively minor point and I would understand if this discussion were precluded by the word count.

We thank the reviewer for their kind words and appreciate their effort in reviewing our manuscript. We have added additional discussion of VAF cutoffs to the introduction.

REVIEWERS' COMMENTS:

Reviewer #1 (Remarks to the Author):

I thank the authors for careful evaluation of the reviewer comments. The responses were overall excellent. A few remaining comments below.

REMAINING CONCERNS:

(1) Rebuttal statement: "The major difference between the two studies is the VAF cutoff which was 0.02 for NHL and 0.01 for MM. However, when we use the same VAF cutoff of 0.02, CHIP prevalence in MM is 14% (compared to 29% in the NHL study) as already stated in our results section. Thus, despite any differences in sequencing approaches we believe that CHIP prevalence in MM prior to ASCT is lower than that of NHL."

- Thanks. Please clearly outline these and any other differences in the supplement. For example: the exclusion/inclusion of "non-driver" mutations, minimum supporting reads for an alt allele (more stringency whether it is applied as minimum number of supporting reads or, for example, a statistically principled approach such as MuTect's LOD score will appear to lower the prevalence). It is crucial to not allow false notions of differences in prevalence to permeate the literature.

(2) Data accessibility: fastq data should be made available in dbGaP or other suitable repository with clinical and demographic covariates to the extent that is permissible by patient consent and other regulatory components, if any. A spreadsheet containing mutation calls (subject identifier, chr, pos, ref allele, alt allele, HGVS cDNA, HGVS protein, # alt reads, #ref reads, total depth, LOD score [or other score], VAF) used for the manuscript and clinical/demographic data for positive and negative CHIP cases should be made available as supplementary data. Alongside known mutation data for the tumor itself -- not embedded in the PDF.

Reviewer #2 (Remarks to the Author):

I note the changes made (primarily in response to the other reviewer) and believe that these have strengthened the manuscript. They have also addressed my one minor criticism. I have no further comments to make.

REVIEWERS' COMMENTS:

Reviewer #1 (Remarks to the Author):

I thank the authors for careful evaluation of the reviewer comments. The responses were overall excellent. A few remaining comments below.

REMAINING CONCERNS:

(1) Rebuttal statement: "The major difference between the two studies is the VAF cutoff which was 0.02 for NHL and 0.01 for MM. However, when we use the same VAF cutoff of 0.02, CHIP prevalence in MM is 14% (compared to 29% in the NHL study) as already stated in our results section. Thus, despite any differences in sequencing approaches we believe that CHIP prevalence in MM prior to ASCT is lower than that of NHL."

- Thanks. Please clearly outline these and any other differences in the supplement. For example: the exclusion/inclusion of "non-driver" mutations, minimum supporting reads for an alt allele (more stringency whether it is applied as minimum number of supporting reads or, for example, a statistically principled approach such as MuTect's LOD score will appear to lower the prevalence). It is crucial to not allow false notions of differences in prevalence to permeate the literature.

We have further outlined our algorithm for calling mutations as "somatic driver mutations" by adding the highlighted statements to the supplementary appendix, under "computational analyses" on page 4, paragraph 4:

Variants were classified as pathogenic driver mutations based on mutation type, position, and frequency in published reports¹⁷⁻¹⁹ and public databases²⁰. The set of rules to consider a queried mutation as a "driver" mutation is outlined in supplementary table 2. The minimum number of alternate reads we chose to accept or reject to call a variant according to MuTect2 was not a hard cutoff but rather one determined as a function of the read depth, strand biases and contamination at a given site⁶. Consequently, the lowest number of accepted alternate reads was 4. Variants with allele fraction less than 0.01 were excluded and, except for *DNMT3A*, *TET2*, *ASXL1*, *PPM1D*, *TP53*, *JAK2*, *SF3B1*, *SRSF2*, mutations with VAF above 0.35 were also excluded since these often represent germline polymorphisms. To further confirm that our called mutations are somatic drivers, we excluded single nucleotide variant (SNV) mutations with a TLOD score below 6.3 (via MuTect2) and Insertion-Deletion (Indel) mutations with a QSI_NT score below 30 (via Strelka).

(2) Data accessibility: fastq data should be made available in dbGaP or other suitable repository with clinical and demographic covariates to the extent that is permissible by patient consent and other regulatory components, if any. A spreadsheet containing mutation calls (subject identifier, chr, pos, ref allele, alt allele, HGVS cDNA, HGVS protein, # alt reads, #ref reads, total depth, LOD score [or other score], VAF) used for the manuscript and clinical/demographic data for positive and negative CHIP cases should be made available as supplementary data. Alongside known mutation data for the tumor itself -- not embedded in the PDF.

We are in the process of updating our previously created study on dbGaP, Multiple Myeloma Genomics Study (MMGS) with accession number phs001323, to include the data of this study. Both sequencing and phenotypic (demographic/clinical) information on all 629 patients will be available on dbGaP within a few weeks.

Spreadsheets containing all mutation calls already exist in the supplementary appendix (Supplementary table 3 and 5). We have added more columns to the spreadsheets to include # alt reads, #ref reads, total depth, TLOD score for SNVs and the QSI_NT score for Indels.

As already indicated in the manuscript, on page 6 paragraph 2, tumor samples were not available for sequencing. Thus, no mutation data exists for the tumor. This is why we utilized ultra-low pass whole genome sequencing to confirm the absence or low level of tumor cells in our samples.

Reviewer #2 (Remarks to the Author):

I note the changes made (primarily in response to the other reviewer) and believe that these have strengthened the manuscript. They have also addressed my one minor criticism. I have no further comments to make.